# Development and validation of an interpretable machine learning model for predicting left atrial thrombus or spontaneous echo contrast in non-valvular atrial fibrillation patients

**Chaoqun Huang, Shangzhi Shu, Miaomiao Zhou[☯], Zhenming Sun[☯], Shuyan Li[ID]***

Department of Cardiovascular Medicine, The First Bethune Hospital of Jilin University, Changchun, Jilin Province, China

☯ These authors contributed equally to this work.

\* li_sy@jlu.edu.cn

**Data Availability Statement:** Data are available from the Atrial Fibrillation Database of the Department of Cardiology, The First Hospital of

## Abstract

### Purpose

Left atrial thrombus or spontaneous echo contrast (LAT/SEC) are widely recognized as significant contributors to cardiogenic embolism in non-valvular atrial fibrillation (NVAF). This study aimed to construct and validate an interpretable predictive model of LAT/SEC risk in NVAF patients using machine learning (ML) methods.

### Methods

Electronic medical records (EMR) data of consecutive NVAF patients scheduled for catheter ablation at the First Hospital of Jilin University from October 1, 2022, to February 1, 2024, were analyzed. A retrospective study of 1,222 NVAF patients was conducted. Nine ML algorithms combined with demographic, clinical, and laboratory data were applied to develop prediction models for LAT/SEC in NVAF patients. Feature selection was performed using the least absolute shrinkage and selection operator (LASSO) and multivariate logistic regression. Multiple ML classification models were integrated to identify the optimal model, and Shapley Additive exPlanations (SHAP) interpretation was utilized for personalized risk assessment. Diagnostic performances of the optimal model and the CHA$_2$DS$_2$-VASc scoring system for predicting LAT/SEC risk in NVAF were compared.

### Results

Among 1,078 patients included, the incidence of LAT/SEC was 10.02%. Six independent predictors, including age, non-paroxysmal AF, diabetes, ischemic stroke or thromboembolism (IS/TE), hyperuricemia, and left atrial diameter (LAD), were identified as the most valuable features. The logistic classification model exhibited the best performance with an area under the receiver operating characteristic curve (AUC) of 0.850, accuracy of 0.812,

Jilin University (Database Administrator Contact: 575407532@qq.com) for researchers who meet the criteria for access to confidential data.

**Funding:** 2. The current work was funded by the National Natural Science Foundation of China (Project No. 82070524). The funders had no role in study design, data collection and analysis, decision to publish, or preparation of the manuscript.

**Competing interests:** The authors have declared that no competing interests exist.

sensitivity of 0.818, and specificity of 0.780 in the test set. SHAP analysis revealed the contribution of explanatory variables to the model and their relationship with LAT/SEC occurrence. The logistic regression model significantly outperformed the $CHA_2DS_2$-VASc scoring system, with AUCs of 0.831 and 0.650, respectively (Z = 7.175, P < 0.001).

## Conclusions

ML proves to be a reliable tool for predicting LAT/SEC risk in NVAF patients. The constructed logistic regression model, along with SHAP interpretation, may serve as a clinically useful tool for identifying high-risk NVAF patients. This enables targeted diagnostic evaluations and the development of personalized treatment strategies based on the findings.

## Introduction

Atrial fibrillation (AF) is one of the most prevalent arrhythmias, predisposing individuals to thromboembolic events, heart failure, and hospitalizations, while concurrently diminishing life quality, exercise capacity [1]. The prevalence of AF is on the rise, with 33.5 million in 2010 [2], that rose to 37.6 million in 2017 [3], and is projected to double by number in the year 2050 [4].

Systemic embolism, particularly stroke, stands out as a pivotal complication of AF, with AF patients facing a fourfold higher risk of ischemic stroke [5]. While the $CHA_2DS_2$-VASc score is extensively utilized to stratify stroke risk in NVAF patients [6], its correlation with left atrial thrombus (LAT) formation exhibits limitations. Left atrial appendage thrombus is part of LAT. The unique anatomical features and functional properties of the left atrial appendage render it the primary site for thrombus formation. LAT and SEC are recognized as significant contributors to cardiogenic embolism in NVAF [7]. However, transesophageal echocardiography (TEE), the gold standard for detecting LAT and SEC, is semi-invasive by nature, with rare but inherent risks and leads to patient discomfort and necessitates some form of sedation. And it demands specialized skills for accurate performance and interpretation [8]. Hence, a potentially non-invasive and efficacious method capable of identifying LAT/SEC would hold substantial clinical value.

ML represents an emerging frontier in medicine, embodying a potent suite of algorithms adept at representing, adapting to, learning from, predicting, and analyzing data. ML is poised as the future of biomedical research, personalized medicine, and computer-aided diagnosis [9, 10]. While several studies have prognosticated LAT/SEC risk in NVAF [11–16], there is limited research on utilizing machine learning to develop predictive models for thrombus formation in AF patients.

Therefore, this study pursues three primary objectives: firstly, to pinpoint the important variables for predicting LAT/SEC in NVAF patients; secondly, to identify the optimal performing ML model for predicting LAT/SEC, utilizing SHAP values to quantitatively visualize the relationships between risk factors and outcomes; and finally, to compare the ML-based prediction model with the conventional $CHA_2DS_2$-VASc scoring system.

## Materials and methods

### Materials

**Subjects.**    A total of 1,222 NVAF patients were scheduled to undergo catheter ablation in the Department of Cardiology, the First Hospital of Jilin University from October 1, 2022 to

February 1, 2024. Data for this study was obtained June 1, 2024. Information about identifying individual participants was concealed during or after data collection.

**Inclusion criteria.**   The inclusion criteria were as follows: (1) NVAF who underwent TEE; (2) informed consent and voluntary participation in the study; and (3) having complete clinical data.

**Exclusion standards.**   The exclusion criteria were as follows: (1) patients who were unable to cooperate, unwilling to participate; (2) patients with rheumatic heart disease and severe valvular heart disease; (3) patients with permanent AF; (4) patients unable to tolerate at least 3 month of oral anticoagulation therapy post catheter ablation; (5) patients who did not undergo TEE; (6) patients with incomplete clinical data.

## Methods

**Grouping methods and diagnosis of LAT and SEC.**   Patients were divided into two groups according to the presence or absence of LAT or SEC. LAT was defined as an echodense mass with tissue characteristics distinct from the left atrial endocardial wall, while SEC was characterized by an echogenic, swirling blood flow pattern at standard gain settings during the cardiac cycle [17].

**Study indicators.**   Demographic, clinical, and laboratory data were extracted from the EMR.

1. Demographic data included age, sex, body mass index (BMI), AF type (paroxysmal or non-paroxysmal), hyperuricemia, history of hypertension, diabetes, ischemic stroke or thromboembolism (IS/TE), coronary heart disease (CHD), hypertrophic cardiomyopathy (HCM), presence of tumors, hypothyroidism, hyperthyroidism, and $CHA_2DS_2$-VASc score. Paroxysmal AF was defined as AF episodes terminating within 7 days (either spontaneously or with medical intervention), while non-paroxysmal AF refers to other types of AF [6]. Hyperuricemia was defined as uric acid levels exceeding 420 umol/L in men and 360 umol/L in women [18].

2. Laboratory parameters included creatinine (Cr), alanine aminotransferase (ALT), aspartate aminotransferase (AST), serum albumin (ALB), fasting blood glucose (FBG), total cholesterol (TC), triglycerides (TG), low-density lipoprotein cholesterol (LDL-c), high-density lipoprotein cholesterol (HDL-c), lymphocyte count (LY), monocyte count (MO), red blood cell count (RBC), hemoglobin (HGB), mean corpuscular volume (MCV), platelet count (PLT), and mean platelet volume (MPV).

3. Echocardiographic parameters encompassed left ventricular ejection fraction (LVEF), left atrial diameter (LAD), left ventricular end-diastolic diameter (LVDD), mitral regurgitation area (MRA), tricuspid regurgitation area (TRA), interventricular septum thickness (IVST), and left ventricular posterior wall thickness (LVPWT).

**Feature selection.**   Initially, R software (glmnet 4.1.8) was employed to perform LASSO regression, a widely utilized technique for feature selection. LASSO regression constructs a penalty function that compresses some regression coefficients, enforcing the sum of the absolute values of coefficients to be less than a predetermined threshold while setting some coefficients to zero, thereby achieving model refinement. LASSO regression retains the advantage of subset shrinkage as a biased estimator, particularly effective for datasets with complex covariance structures. This algorithm utilizes a 10-fold cross-validation approach to automatically eliminate features with zero coefficients. Subsequently, the results of LASSO regression

analysis were utilized to conduct multivariate logistic regression analysis, ultimately identifying significant factors with P < 0.05.

**ML model establishment and development.** Nine algorithms were utilized to develop and compare prediction models. The characteristic factors that were selected based on LASSO and multivariate logistic regression were used in the prediction models. The ML models, including eXtreme Gradient Boosting (XGBoost), Logistic Regression, Light Gradient Boosting Machine (LightGBM), RandomForest, Adaptive Boosting (AdaBoost), DecisionTree, Gradient Boosting (GBDT), Gaussian Naïve Bayes (GNB), and Complement Naïve Bayes (CNB), were constructed using Python (scikit-learn 0.22.1, xgboost 1.2.1, lightgbm 3.2.1). A bootstrap resampling technique was employed to train and validate the classification of the ML models. The patients were randomly divided into training and test sets (7:3). The validation dataset was utilized to evaluate and compare the performance of each model. The AUC, accuracy, sensitivity, specifcity, positive predictive value (PPV), negative predictive value (NPV), and F1 scores were used to evaluate the ability of the model to predict LAT/SEC in patients with NVAF. Calibration curves were employed to evaluate the predictive power of the model, and a comprehensive assessment of the predictive model was conducted to validate its utility in decision support or broader simulation modeling.

**Model optimization and evaluation.** To ensure model stability, 10-fold cross-validation was employed to evaluate the predictive ability of the model. The training set was randomly divided into 10 groups, with 9 groups used for training in each iteration of the 10-fold cross-validation, and the remaining group designated as the validation set. During each training iteration, a 30% subset was randomly sampled from the training data to assess model performance. Subsequently, model discrimination was quantified using receiver operating characteristic (ROC) curve analysis, and predictive accuracy was evaluated using the obtained AUCs and calibration. Decision curve analysis (DCA) was utilized to estimate clinical utility and net benefit. Feature importance was assessed using SHAP, with higher absolute SHAP values indicating features that had the greatest impact on the model's prediction score. Additionally, SHAP was employed to calculate prediction performance for an individual sample.

## Bias control

The process of data collection should strictly ensure the comprehensiveness, accuracy, non-duplication, and clear definition of the collected data.

## Study size

According to predictive research, the effective sample size is determined by the number of outcome events, which should be 5–10 times the number of included variables. This study includes 23 observational indicators, thus the estimated number of outcome events should be at least 115 cases. Preliminary trials indicate that the incidence of LAT/SEC is approximately 11%. Therefore, the required sample size for the training set should be at least 1045 cases.

## Statistical analysis

Continuous variables were presented as mean ± standard deviation or median with interquartile range and analyzed using the unpaired t-test or the Mann-Whitney U test, as appropriate. Categorical variables were expressed as absolute numbers (n) and relative frequencies (%) and analyzed using the Chi-squared test. Bilateral p-values less than 0.05 were considered statistically significant. Statistical analysis was conducted using SPSS (version 25.0), R (version 4.2.3), Python (version 3.11.4), and MedCalc (version 22.009).

### Ethics declarations

The study was approved by the Review Ethics Committee of the First Bethune Hospital of Jilin University (approval number: 2024–665), and informed consent was waived due to the retrospective nature of the study.

## Results

### Comparison of baseline data

During the study period, 1,222 consecutive NVAF patients were enrolled, and 144 patients were excluded due to incomplete data. Ultimately, a total of 1078 patients were included in this study. The demographic, clinical, and laboratory data of the patients are summarized in Table 1. Among them, 15 patients had LAT and 93 had SEC, resulting in an overall incidence of LAT/SEC of 10.02% (108/1078). The majority of participants were male (63.27%), and 43.6% had non-paroxysmal AF. The median age was 62 years, the median LVEF was 62%, and the median $CHA_2DS_2$-VASc score was 2. Compared to the control group, patients with LAT/SEC were older, had a higher prevalence of non-paroxysmal AF, diabetes, hyperuricemia, history of IS/TE, and tumor, and exhibited lower ABL, LVEF, and higher AST, ALT, FBG, Crea, HGB, as well as larger MCV, MPV, LAD, LVDD, MRA, TRA, and IVST. However, other anthropometric, biochemical, and clinical parameters did not show significant differences between the two groups. Notably, in patients with HCM or cardiac amyloidosis in conjunction with AF, the risk of stroke is significantly increased. Current studies recommend that such patients should routinely receive anticoagulation therapy, regardless of their $CHA_2DS_2$-VASc score [6]. However, in this study, the prevalence of HCM did not differ significantly between the two groups, which may be due to the small sample size of HCM patients, leading to a lack of statistical significance. Additionally, patients with cardiac amyloidosis were not mentioned in this study due to the unavailability of relevant data.

### Feature selection and comparison of multiple classification models

We initially analyzed 37 variables excluding the stroke score. Among them, 23 variables with a P-value of less than 0.1 were selected based on the univariate analysis. These variables included age, BMI, non-paroxysmal AF, hypertension, diabetes, IS/TE, tumor, hyperuricemia, ABL, AST, ALT, FBG, Crea, HGB, MCV, MPV, LVEF, LAD, LVDD, MRA, TRA, IVST, and LVPWT. LASSO regression, a technique that can compress variable coefficients to prevent overfitting and resolve severe collinearity issues, was employed. The LASSO regression analysis (Fig 1) revealed that, at a lambda value with minimum mean square error of 0.007, the initial 23 independent variables were reduced to 13. These 13 variables included age, non-paroxysmal AF, diabetes, IS/TE, tumor, hyperuricemia, AST, ALT, MCV, MPV, LVEF, LAD, and IVST. Subsequently, to further control for the influence of confounding factors, the aforementioned 13 independent variables underwent multivariate logistic regression analysis. Ultimately, only age, non-paroxysmal AF, diabetes, IS/TE, hyperuricemia, and LAD were determined as characteristic factors (p < 0.05), as demonstrated in Table 2.

### Comprehensive analysis of classified multi-model

The performance of the 9 ML classification models in predicting LAT/SEC risk in both the training and validation sets was compared and summarized in Table 3 and Fig 2. Upon a comprehensive evaluation utilizing multiple indicators, it was observed that the logistic regression model exhibited the most robust performance in predicting LAT/SEC among NVAF patients. Fig 2a and 2b depict the comparison of ROC curves for different ML models in both the

**Table 1. Comparison of baseline characteristics between the LAT/SEC and control cohorts.**

| Variable | Total (n = 1078) | Control group (n = 970) | LAT/SEC group (n = 108) | P-value |
|---|---|---|---|---|
| Male sex, n(%) | 682(63.27) | 607(62.58) | 75(69.45) | 0.160 |
| Age, year | 62.00[56.00–69.00] | 62.00[56.00–69.00] | 66.00[59.00–70.00] | 0.007 |
| BMI, kg/m$^2$ | 25.56[23.26–27.78] | 25.51[23.15–27.72] | 26.04[23.69–28.58] | 0.064 |
| Non-paroxysmal AF, n(%) | 470(43.60) | 381(39.28) | 89(82.41) | <0.001 |
| CHA$_2$DS$_2$-VASc | 2.00[1.00,3.00] | 2.00[1.00,3.00] | 3.00[2.00,4.00] | <0.001 |
| Hypertension, n(%) | 552(51.21) | 488(50.31) | 64(59.26) | 0.078 |
| Diabetes, n(%) | 277(25.70) | 237(24.43) | 40(37.04) | 0.004 |
| CHD, n(%) | 238(22.08) | 219(22.58) | 19(17.59) | 0.236 |
| IS/TE, n(%) | 127(11.78) | 98(10.10) | 29(26.85) | <0.001 |
| HCM, n(%) | 16(1.48) | 13(1.34) | 3(2.78) | 0.241 |
| prosthetic heart valves, n(%) | 12(1.11) | 10(1.03) | 2(1.85) | 0.432 |
| Tumor, n(%) | 49(4.55) | 39(4.02) | 10(9.26) | 0.013 |
| Hyperthyroidism, n(%) | 31(2.88) | 28(2.89) | 3(2.78) | 0.949 |
| Hypothyroidism, n(%) | 29(2.69) | 27(2.78) | 2(1.85) | 0.570 |
| Hyperuricemia, n(%) | 297(27.55) | 248(25.57) | 49(45.37) | <0.001 |
| ABL, g/L | 38.90[36.90–41.20] | 39.00[37.10–41.20] | 38.10[35.80–41.00] | 0.032 |
| AST, U/L | 20.90[17.30–26.00] | 20.80[17.10–25.50] | 24.00[18.70–30.20] | <0.001 |
| ALT, U/L | 20.10[14.5029.60] | 19.80[14.30–29.10] | 25.20[16.70–36.90] | <0.001 |
| FBG, mmol/L | 5.32[4.86–6.03] | 5.30[4.85–5.97] | 5.55[4.92–6.51] | 0.018 |
| TC, mmol/L | 4.28[3.58–4.95] | 4.28[3.58–4.96] | 4.37[3.61–4.86] | 0.797 |
| TG, mmol/L | 1.33[0.97–1.90] | 1.33[0.97–1.91] | 1.50[1.05–1.88] | 0.582 |
| HDL-c, mmol/L | 1.04[0.90–1.21] | 1.04[0.90–1.20] | 1.05[0.95–1.21] | 0.235 |
| LDL-c, mmol/L | 2.66[2.14–3.20] | 2.66[2.14–3.19] | 2.70[2.12–3.21] | 0.908 |
| Crea, umol/L | 72.60[61.50–85.20] | 72.30[61.30–84.80] | 76.40[65.00–88.80] | 0.014 |
| LY, x10$^9$/L | 1.90[1.51–2.39] | 1.91[1.51–2.38] | 1.85[1.41–2.40] | 0.658 |
| MO, x10$^9$/L | 0.48[0.39–0.60] | 0.48[0.39–0.60] | 0.50[0.40–0.63] | 0.127 |
| RBC, x10$^{12}$/L | 4.82[4.46–5.22] | 4.81[4.46–5.22] | 4.96[4.51–5.23] | 0.156 |
| HGB, g/L | 149.11±16.81 | 148.73±16.75 | 152.48±16.94 | 0.028 |
| MCV, fL | 91.10[88.40–94.00] | 90.90[88.30–93.90] | 92.30[89.70–94.10] | 0.019 |
| PLT, x10$^9$/L | 220.00[183.0–260.0] | 220.00[183.0–260.0] | 216.0[173.0–255.0] | 0.286 |
| MPV, fL | 10.30[9.70–10.90] | 10.30[9.70–10.90] | 10.50[10.10–11.20] | 0.009 |
| LVEF, % | 62.00[57.00–64.00] | 62.00[58.00–64.00] | 59.00[50.00–61.00] | <0.001 |
| ≥50% | 992(92.02) | 907(93.51) | 85(78.70) | |
| <50% | 86(7.98) | 63(6.49) | 23(21.30) | <0.001 |
| LAD, mm | 39.00[35.00–43.00] | 39.00[35.00–42.00] | 45.00[42.00–48.00] | <0.001 |
| LVDD, mm | 48.00[46.00–52.00] | 48.00[45.00–51.00] | 50.00[46.00–54.00] | 0.003 |
| MRA, cm$^2$ | 2.00[0.00–3.90] | 2.00[0.00–3.60] | 3.50[2.000–5.50] | <0.001 |
| TRA, cm$^2$ | 2.10[0.00–4.00] | 2.00[0.00–3.80] | 3.80[2.30–6.20] | <0.001 |
| IVST, mm | 9.00[8.00–10.00] | 9.00[8.00–10.00] | 10.00[9.00–10.00] | 0.002 |
| LVPWT, mm | 9.00[8.00–9.00] | 9.00[8.00–9.00] | 9.00[8.00–10.00] | 0.068 |

training and validation sets. Notably, DecisionTree was found to be more susceptible to overfitting, while logistic regression demonstrated relatively stable performance. Calibration curves (Fig 2c) were constructed to assess the accuracy of the models. Additionally, the forest plot (Fig 2d) illustrates the ROC results for LAT/SEC prediction by each model, with error bars indicating the mean and standard deviation of the ROC.

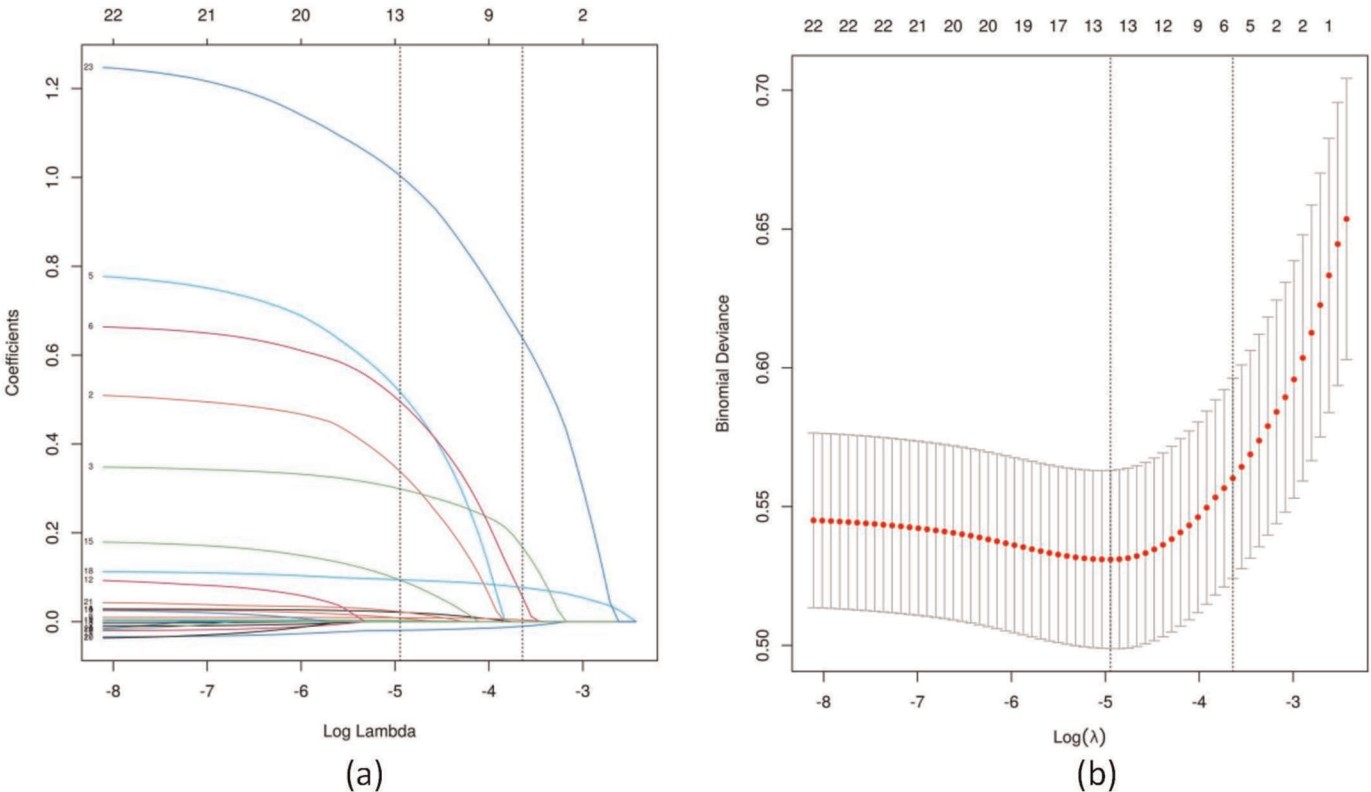

**Fig 1. LASSO regression analysis. (a)** The use of 10-fold cross-validation to draw vertical lines at selected values, where the optimal lambda produces 5 nonzero coefficients. **(b)** In the LASSO model, the coefficient profiles of 23 texture features were drawn from the log (λ) sequence. Vertical dotted lines are drawn at the minimum mean square error (λ = 0.007) and the standard error of the minimum distance (λ = 0.026).

**Table 2. Multivariate logistic regression analysis.**

| Variable | R | SE | Z | p | OR (95% CI) |
|---|---|---|---|---|---|
| Age | 0.032 | 0.014 | 2.3 | 0.021 | 1.032 (1.005–1.061) |
| AST | 0.011 | 0.006 | 1.728 | 0.084 | 1.011 (0.998–1.024) |
| ALT | 0.004 | 0.006 | 0.741 | 0.458 | 1.004 (0.992–1.016) |
| MPV | 0.183 | 0.134 | 1.366 | 0.172 | 1.200 (0.923–1.561) |
| MCV | 0.028 | 0.026 | 1.069 | 0.285 | 1.028 (0.977–1.082) |
| LVEF | -0.023 | 0.013 | -1.772 | 0.076 | 0.977 (0.953–1.003) |
| LAD | 0.104 | 0.022 | 4.835 | <0.001 | 1.110 (1.064–1.159) |
| IVST | 0.046 | 0.05 | 0.925 | 0.355 | 1.047 (0.945–1.154) |
| Diabetes | 0.559 | 0.253 | 2.206 | 0.027 | 1.748 (1.059–2.864) |
| IS/TE | 0.709 | 0.28 | 2.531 | 0.011 | 2.033 (1.161–3.493) |
| Tumour | 0.771 | 0.457 | 1.687 | 0.092 | 2.163 (0.843–5.134) |
| Hyperuricemia | 0.705 | 0.242 | 2.908 | 0.004 | 2.023 (1.256–3.256) |
| Non-paroxysmal AF | 1.206 | 0.298 | 4.051 | <0.001 | 3.338 (1.894–6.115) |
| (Intercept) | -13.894 | 3.164 | -4.391 | 0.0 | 0 (-) |

R regression coefficient, SE standard error, OR odds ratio, CI confidence interval.

**Table 3. Predictive performance of 9 ML algorithms in training and validation sets for LAT/SEC in NVAF patients.**

| Models | AUC | Accuracy | Sensitivity | Specificity | PPV | NPV | F1 score |
|---|---|---|---|---|---|---|---|
| **Training set** | | | | | | | |
| XGBoost | 0.996 | 0.971 | 0.985 | 0.970 | 0.788 | 0.997 | 0.874 |
| logistic | 0.833 | 0.774 | 0.786 | 0.773 | 0.278 | 0.969 | 0.410 |
| LightGBM | 0.998 | 0.976 | 0.998 | 0.974 | 0.816 | 0.999 | 0.897 |
| RandomForest | 0.999 | 0.986 | 0.995 | 0.984 | 0.892 | 0.998 | 0.940 |
| AdaBoost | 0.885 | 0.759 | 0.925 | 0.740 | 0.283 | 0.987 | 0.433 |
| DecisionTree | 1.000 | 0.992 | 1.000 | 0.99 | 0.954 | 0.996 | 0.976 |
| GBDT | 0.946 | 0.861 | 0.919 | 0.855 | 0.413 | 0.988 | 0.569 |
| GNB | 0.821 | 0.739 | 0.796 | 0.733 | 0.250 | 0.969 | 0.379 |
| CNB | 0.695 | 0.475 | 0.858 | 0.434 | 0.143 | 0.963 | 0.245 |
| **Validation set** | | | | | | | |
| XGBoost | 0.771 | 0.839 | 0.908 | 0.608 | 0.259 | 0.922 | 0.39 |
| logistic | 0.830 | 0.770 | 0.881 | 0.736 | 0.274 | 0.966 | 0.414 |
| LightGBM | 0.782 | 0.834 | 0.890 | 0.665 | 0.207 | 0.915 | 0.326 |
| RandomForest | 0.717 | 0.841 | 0.715 | 0.733 | 0.220 | 0.915 | 0.322 |
| AdaBoost | 0.790 | 0.732 | 0.826 | 0.742 | 0.241 | 0.964 | 0.371 |
| DecisionTree | 0.561 | 0.841 | 0.214 | 0.914 | 0.219 | 0.911 | 0.210 |
| GBDT | 0.813 | 0.809 | 0.881 | 0.712 | 0.311 | 0.954 | 0.454 |
| GNB | 0.820 | 0.725 | 0.882 | 0.727 | 0.238 | 0.964 | 0.374 |
| CNB | 0.689 | 0.473 | 0.799 | 0.586 | 0.136 | 0.956 | 0.231 |

AUC area under the curve, PPV positive predictive value, NPV negative predictive value, XGBoost extreme gradient boosting, LightGBM light gradient boosting machine, AdBoost adaptive boosting, GBDT GradientBoosting, GNB Gaussian Naïve Bayes, CNB Complement Naïve Bayes.

## Optimal model construction and evaluation

Logistic regression analysis and 10-fold cross-validation were conducted. The results demonstrated that the average AUC of the training set was 0.825 (95% CI 0.780–0.871), the average AUC of the validation set was 0.814 (95% CI 0.682–0.943), and the AUC of the test set was 0.850 (95% CI 0.780–0.920) (Table 4 and Fig 3a–3c). Meanwhile, we have presented the performance of $CHA_2DS_2$-VASc in predicting LAT/SEC across the training, testing, and validation sets. Considering that the performance of the validation set in terms of the AUC index was slightly lower than that of the test set, or the difference was less than 10%, the model fitting was considered successful. Additionally, the learning curve illustrated that both the training and validation sets exhibited strong fitting and high stability (Fig 3d). Furthermore, calibration plots (Fig 3e) were utilized to assess the accuracy of the model, revealing excellent concordance between the predicted probabilities of the logistic regression model and observed LAT/SEC rates. Subsequent construction of the Decision Curve Analysis (DCA) for this model in our study (Fig 3f) suggested that the ML model provided a greater net benefit compared to a treat-all or treat-none strategy, with a risk threshold ranging from approximately 5% to 67%. Additionally, the KS statistic shows that when the absolute KS value reaches its maximum (0.598), the predicted probability is 0.104 (S1 Fig). This indicates strong model discrimination, with patients at higher risk of LAT/SEC when the predicted score exceeds 0.104. These findings underscored the suitability of the logistic regression model for the classification modeling task of the dataset.

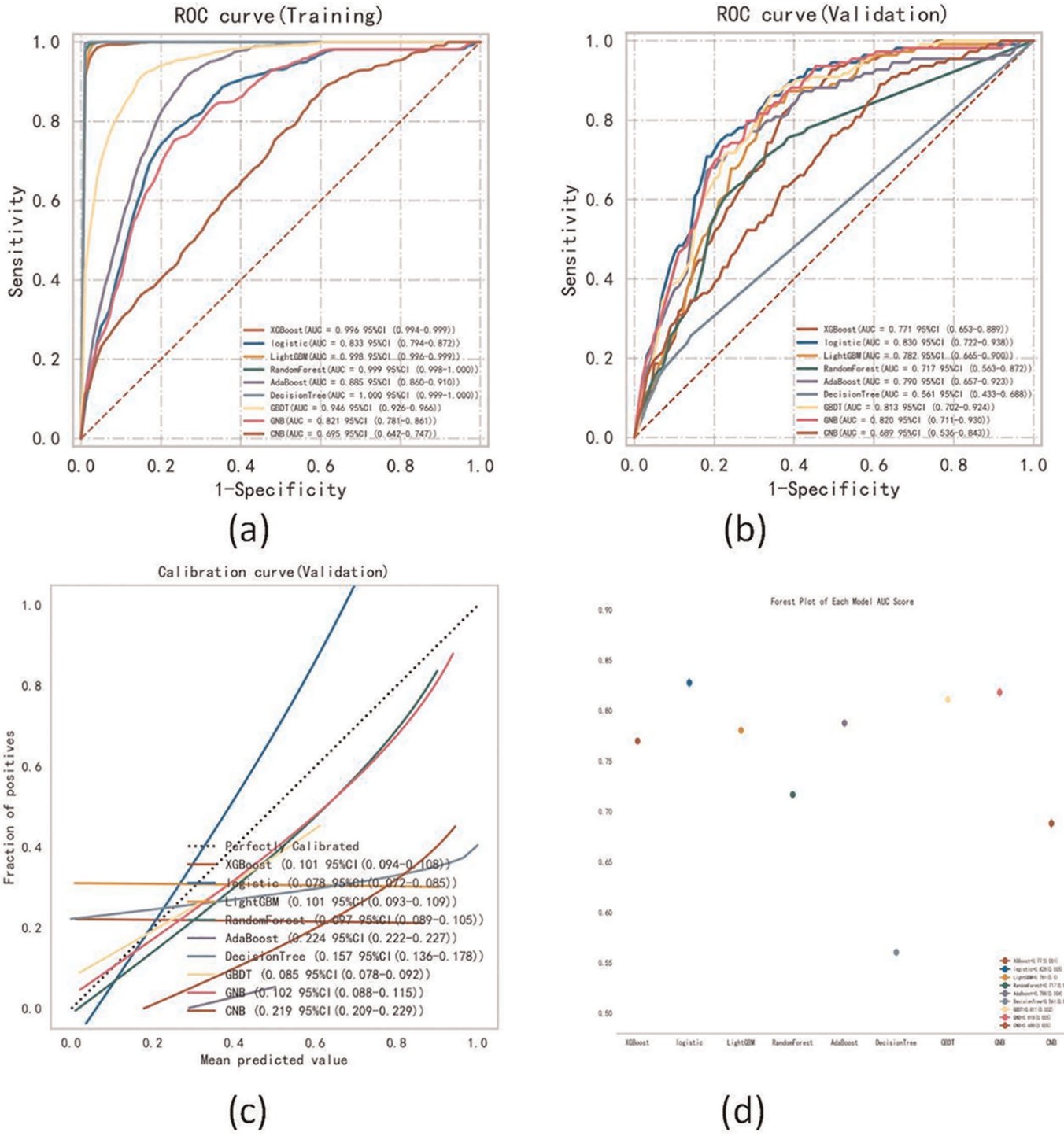

**Fig 2. Comprehensive analysis of ML models. (a)** ROC curve analysis of ML algorithms for the prediction of LAT/SEC in the training set. **(b)** ROC curve analysis of ML algorithms for the prediction of LAT/SEC in the validation set. **(c)** Calibration plots for predicting LAT/SEC in NVAF patients using various models. **(d)** Forest Plot of each model AUC score.

**Table 4. Diagnostic performance of the logistic regression model for the prediction of LAT/SEC risk in NVAF.**

| Models | AUC | Cutoff | Accuracy | Sensitivity | Specificity | PPV | NPV | F1 score |
|---|---|---|---|---|---|---|---|---|
| **ML model** | | | | | | | | |
| Training set | 0.825 | 0.103 | 0.742 | 0.818 | 0.734 | 0.255 | 0.972 | 0.387 |
| Validation set | 0.814 | 0.103 | 0.731 | 0.907 | 0.732 | 0.247 | 0.963 | 0.383 |
| Test set | 0.850 | 0.114 | 0.812 | 0.818 | 0.780 | 0.321 | 0.967 | 0.461 |
| **CHA$_2$DS$_2$-VASc** | | | | | | | | |
| Training set | 0.637 | 0.116 | 0.778 | 0.342 | 0.827 | 0.187 | 0.919 | 0.237 |
| Validation set | 0.631 | 0.116 | 0.780 | 0.354 | 0.826 | 0.178 | 0.923 | 0.229 |
| Test set | 0.679 | 0.118 | 0.809 | 0.333 | 0.863 | 0.216 | 0.919 | 0.262 |

### Interpretation of the model using SHAP

To elucidate the predictive influence of selected variables, we employed SHAP to illustrate their impact on the formation of LAT/SEC within the model. In Fig 4a, the six most significant features in our model are depicted. Each feature's importance is represented by colored dots, with red indicating high-risk values and blue representing low-risk values. Non-paroxysmal AF, diabetes, IS/TE, hyperuricemia, older age, and larger LAD were associated with increased formation of LAT/SEC in NVAF patients. Fig 4b displays the ranking of these six risk factors based on the average absolute SHAP value, providing insight into their relative importance within the predictive model. We also used SHAP values to demonstrate the relative weightings of each variable in predicting LAT/SEC across the remaining 8 ML models (S2 Fig). Additionally, we present a representative case to illustrate the model's interpretability: an NVAF patient with LAT/SEC exhibiting a high SHAP predictive score (0.17), as depicted in Fig 4c.

### Comparison of diagnostic performances between the optimal model and CHA$_2$DS$_2$-VASc scoring system in predicting the risk of LAT/SEC risk in NVAF

We conducted a comparative analysis of the diagnostic accuracy between the ML-based logistic regression model and the CHA$_2$DS$_2$-VASc scoring system for predicting LAT/SEC risk in NVAF patients (Table 5). The ROC curve of the CHA$_2$DS$_2$-VASc scoring system was 0.650 (95% CI 0.597–0.702), whereas the logistic regression model yielded a higher ROC of 0.831 (95% CI 0.790–0.868). The observed difference between the ROC values was 0.181, which demonstrated statistical significance (Z = 7.175, P < 0.001). Fig 5 illustrates the ROC curves of both models, highlighting the superior performance of the ML-based logistic regression model over the CHA$_2$DS$_2$-VASc scoring system.

### Discussion

While the CHA$_2$DS$_2$-VASc score is commonly utilized for stroke risk stratification in NVAF patients, its efficacy in predicting LAT/SEC, recognized as a major contributor to cardiogenic embolism in NVAF, is limited. Multicenter study showed the C-statistic values of the CHA$_2$DS$_2$-VASc scores concerning LAT/SEC were 0.643 [12], and were 0.650 in this study. Although TEE is the gold standard for detecting LAT and SEC, it requires special skills and is a relatively invasive method. Consequently, there is a pressing need for a non-invasive and effective method to identify NVAF patients at high risk of LAT/SEC, allowing for further diagnostic evaluations, such as TEE, in this population. This would hold significant clinical value.

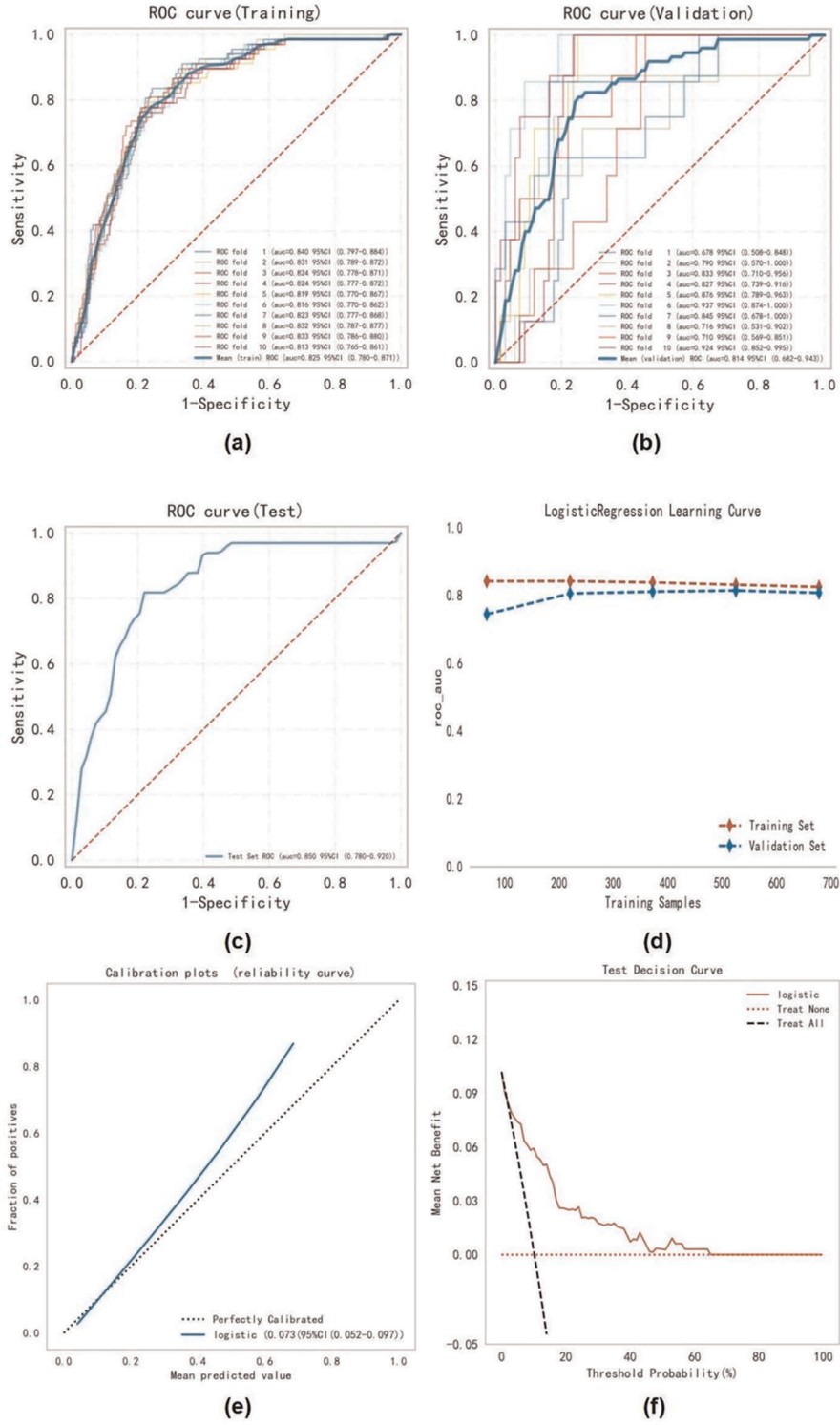

**Fig 3. Logistic regression model evaluation. (a-c)** The ROC curves of logistic regression using the 10-fold cross-validation on the training set (**a**), validation set (**b**), and test set (**c**). (**d**) Machine learning curve. (**e**) calibration plots for logistic regression. (**f**) Decision curve analysis graph showing the net benefit against threshold probabilities based on decisions from model outputs.

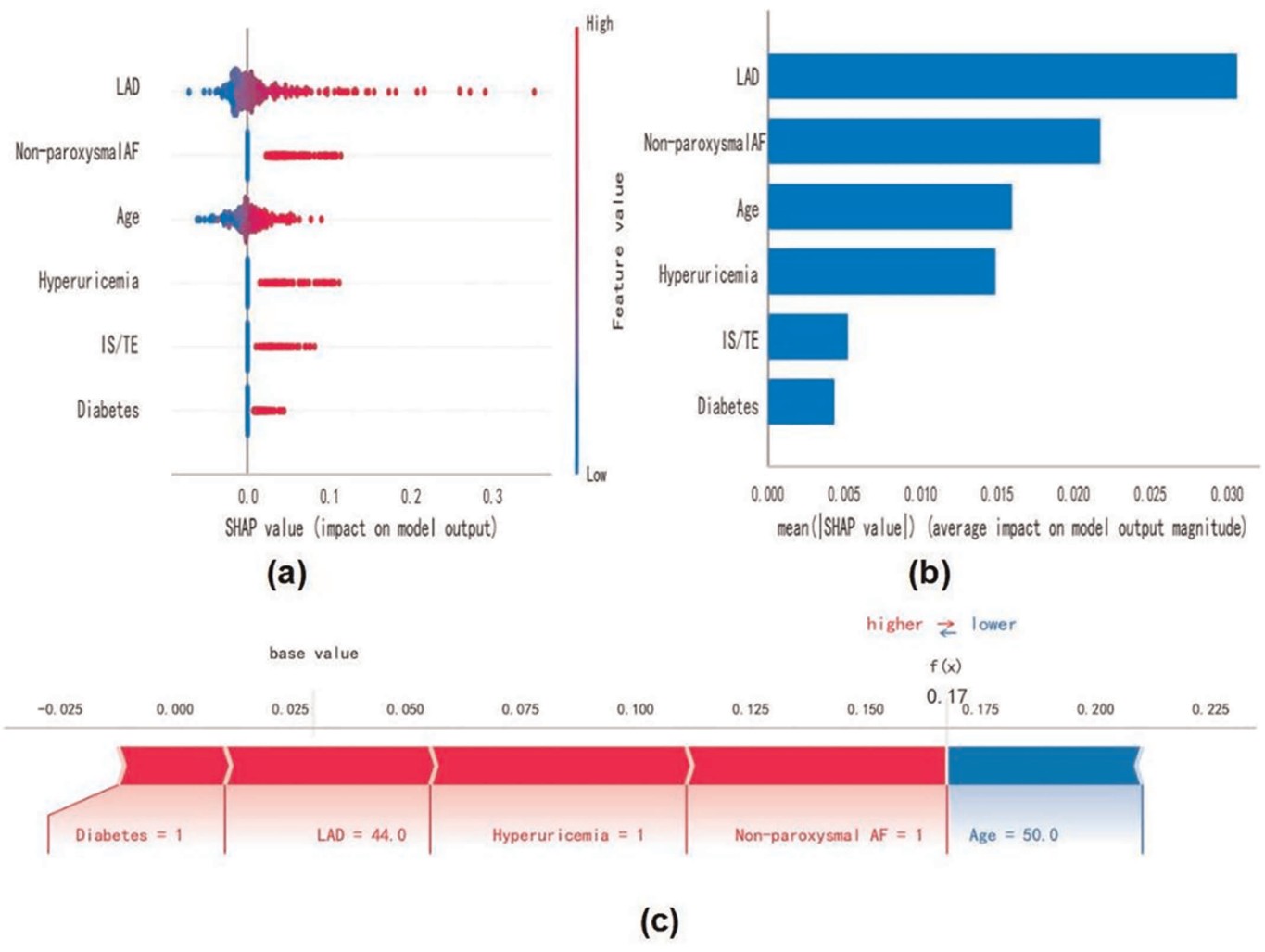

**Fig 4. SHAP analysis of the model. (a)** Feature attributions in SHAP. Each line corresponds to a feature, with SHAP values plotted on the abscissa. Red dots denote higher values, while blue dots represent lower values. **(b)** Importance of variables depicted as bars, indicating their contribution to model predictions. **(c)** SHAP scores elucidate the predicted risk of LAT/SEC in an individual subject.

Previous studies have reported varying incidence rates of LAT/SEC in NVAF patients, ranging from 4.3% to 32.8% [11–13, 15, 19–22]. This discrepancy could stem from inconsistent anticoagulant therapy or potential racial disparities among the enrolled patients. In our study, the prevalence of LAT/SEC was approximately 10%. Unfortunately, recent anticoagulation data for the patients were unavailable. Leveraging LASSO and multivariate logistic regression analyses, we identified age, non-paroxysmal AF, diabetes, IS/TE, hyperuricemia, and LAD as closely associated with LAT/SEC. Among the ML algorithms examined, the logistic regression model exhibited superior performance. We successfully developed a novel ML-

**Table 5. Performance comparison of the proposed ML-based logistic regression model with CHA$_2$DS$_2$-VASc in predicting LAT/SEC risk in NVAF.**

|                      | AUC   | Cutoff | Sensitivity | Specificity |
|----------------------|-------|--------|-------------|-------------|
| CHA$_2$DS$_2$-VASc   | 0.650 | 2.000  | 0.759       | 0.455       |
| Logistic regression  | 0.831 | 0.105  | 0.796       | 0.754       |

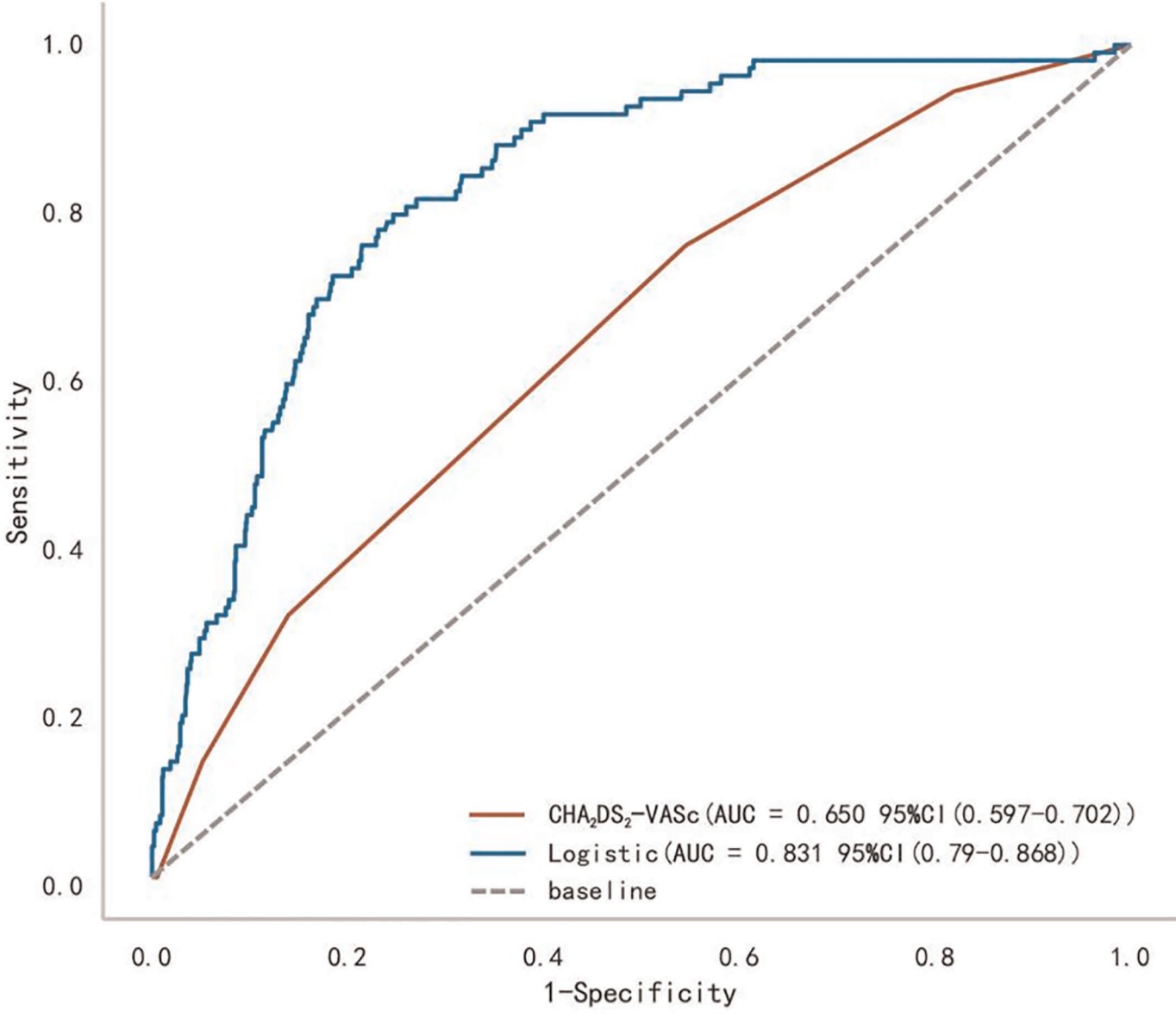

**Fig 5. ROC of the logistic regression model and CHA$_2$DS$_2$-VASc in predicting LAT/SEC risk in NVAF.**

based prediction model for assessing LAT/SEC risk in NVAF patients, further refining its accuracy and clinical validity through automatic parameter adjustment and internal cross-validation. Evaluation using calibration and DCA curves reinforced the model's utility for classifying the dataset. Notably, our ML-based logistic regression model significantly outperformed the CHA$_2$DS$_2$-VASc scoring system. In clinical practice, based on our newly developed predictive model, patients with NVAF who have a high SHAP prediction score—specifically, above 0.104—are at an increased risk of LAT/SEC. Clinicians should recommend that such patients undergo TEE to evaluate the presence of LAT/SEC. Additionally, comprehensive assessments of left atrial and left atrial appendage function, including parameters such as strain or fibrosis, should be performed. Finally, treatment strategies, including anticoagulation, catheter ablation, or LAA closure, should be determined based on the evaluation results.

However, interpreting ML prediction models comprehensively and visually presenting predictive results to clinicians has always been challenging. Additionally, an exploratory analysis of the ML model, as presented in the SHAP value plot, revealed the importance of six variables. These variables ranked in importance from high to low were LAD, non-paroxysmal AF, age, hyperuricemia, IS/TE, and diabetes. Notably, age, IS/TE, and diabetes were included in the $CHA_2DS_2$-VASc score, while LAD, non-paroxysmal AF, and hyperuricemia were not included. In the following, we delve into the impact of LAD, non-paroxysmal AF, and hyperuricemia on LAT and SEC.

Several studies have explored the predictive value of various factors in assessing the risk of LAT or SEC in patients with NVAF. One study identified an enlarged LAD as an independent risk factor for LAT/SEC among NVAF patients with low $CHA_2DS_2$-VASc scores [23]. Similarly, another study noted that the LAT/SEC group exhibited a larger LAD compared to the non-LAT/SEC group among NVAF patients, underscoring the significance of LAD in predicting LAT/SEC [20]. Additionally, a combined predictive model incorporating LAD, LVEF, serum uric acid, and brain natriuretic peptide was developed to evaluate the risk of cardiogenic stroke in NVAF patients, further emphasizing the importance of LAD in risk assessment [24]. Consistent with previous findings, our study identified LAD as the most valuable variable for predicting LAT/SEC, with median LAD measurements of 39 mm and 45 mm in the LAT/SEC group and control group, respectively. These findings highlight the critical role of LAD in predicting LAT/SEC in NVAF patients and underscore the necessity for comprehensive risk assessment models that integrate multiple variables to enhance risk prediction accuracy. However, it is important to recognize the limitations of LAD in accurately assessing left atrial size. More reliable parameters, such as left atrial volume or left atrial volume index, could provide a more accurate assessment and potentially improve the model's reliability. Unfortunately, these data were not available for analysis in this study.

Non-paroxysmal AF emerged as the second most significant predictor variable. Clinically, AF presents in various patterns including first-visit AF, paroxysmal AF, persistent AF, long-term persistent AF, and permanent AF [6]. Clinicians often perceive non-paroxysmal AF to carry a higher stroke risk compared to paroxysmal AF due to prolonged exposure to the embolic-prone state. Notably, the widely accepted $CHA_2DS_2$-VASc scoring system does not account for AF patterns. Recent studies have increasingly emphasized the impact of AF patterns on stroke risk among AF patients. For instance, Vanassche et al. [25] conducted an analysis involving a large cohort of non-anticoagulated AF patients, revealing AF pattern as an independent predictor of stroke risk. They found that the incidence of embolic events was significantly higher in persistent AF (3.0%/year) and permanent AF (4.2%/year) compared to paroxysmal AF (2.1%/year). Similarly, an analysis of the ARISTOTLE database demonstrated significantly lower stroke and embolism rates in paroxysmal AF compared to persistent or permanent AF [26]. Moreover, a meta-analysis involving 99,996 AF patients reported a higher incidence of thromboembolic events among patients with non-paroxysmal AF (HR = 1.38, P < 0.001) [27]. Consistently, several studies have independently concluded that non-paroxysmal AF serves as an independent risk factor for LAT/SEC in NVAF patients [13, 20, 28, 29]. These findings underscore the importance of considering non-paroxysmal AF in predicting both stroke and LAT/SEC.

Hyperuricemia emerged as a more significant predictor than diabetes and IS/TE in our final model. Numerous studies have linked hyperuricemia to endothelial or endocardial dysfunction caused by free radicals, leading to excessive proinflammatory effects [30, 31]. Previous research has established hyperuricemia as an independent risk factor for stroke [32–34]. Furthermore, studies have suggested that hyperuricemia may independently predict and refine the risk of left atrial stasis among NVAF patients, particularly those with a $CHA_2DS_2$-VASc

score < 2 [14]. Consistent with our findings, the prevalence of hyperuricemia was significantly higher in the LAT/SEC group (45.37%) compared to the control group (25.57%). These findings collectively emphasize the importance of hyperuricemia as a critical factor in predicting LAT/SEC in NVAF patients.

Nevertheless, our study is subject to several limitations. Firstly, data regarding anticoagulation in NVAF patients were not available, potentially affecting the accuracy of the final prediction model. Secondly, the sample size was relatively small, and data were collected from a single institution, limiting the generalizability of the findings. Moreover, since our study was not conducted across multiple centers, its applicability to broader populations may be constrained. Additionally, despite achieving high consistency in the repeatability analysis within the training and testing sets, the possibility of segmentation uncertainty introduces potential errors.

## Conclusions

The new model represents a potentially non-invasive and effective approach for predicting the risk of LAT/SEC in NVAF patients. In practical terms, it holds promise for clinical utility by aiding clinicians in identifying high-risk patients with LAT/SEC, thereby enabling facilitating targeted diagnostic evaluations and the formulation of personalized treatment strategies based on the results.

## Supporting information

**S1 Fig. KS statistic plot in predicting LAT/SEC risk.**
(TIF)

**S2 Fig. Feature attributions in SHAP of 9 ML classification models in predicting LAT/SEC risk.**
(TIF)

## Author Contributions

**Data curation:** Miaomiao Zhou, Zhenming Sun.

**Formal analysis:** Chaoqun Huang, Miaomiao Zhou, Zhenming Sun.

**Investigation:** Shangzhi Shu.

**Methodology:** Miaomiao Zhou, Zhenming Sun.

**Software:** Chaoqun Huang, Shangzhi Shu, Zhenming Sun.

**Supervision:** Shuyan Li.

**Validation:** Shangzhi Shu.

**Writing – original draft:** Chaoqun Huang.

**Writing – review & editing:** Shuyan Li.

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
