## [Decision Letter · Decision Letter 0]

9 Oct 2024

PONE-D-24-27654Development and Validation of an Interpretable Machine Learning Model for Predicting Left Atrial Thrombus or Spontaneous Echo Contrast in Non-Valvular Atrial Fibrillation PatientsPLOS ONE

Dear Dr. Li,

Thank you for submitting your manuscript to PLOS ONE. After careful consideration, we feel that it has merit but does not fully meet PLOS ONE’s publication criteria as it currently stands. Therefore, we invite you to submit a revised version of the manuscript that addresses the points raised during the review process.

**ACADEMIC EDITOR: **please address all comments raised by reviewers.

We look forward to receiving your revised manuscript.

Kind regards,

Tom Wang

Academic Editor

PLOS ONE

“The current work was funded by the National Natural Science Foundation of China (Project No. 82070524).”

“The current work was funded by the National Natural Science Foundation of China (Project No. 82070524).”

“The current work was funded by the National Natural Science Foundation of China (Project No. 82070524).”

5. In the online submission form you indicate that your data is not available for proprietary reasons and have provided a contact point for accessing this data. Please note that your current contact point is a co-author on this manuscript. According to our Data Policy, the contact point must not be an author on the manuscript and must be an institutional contact, ideally not an individual. Please revise your data statement to a non-author institutional point of contact, such as a data access or ethics committee, and send this to us via return email. Please also include contact information for the third party organization, and please include the full citation of where the data can be found.

6. PLOS requires an ORCID iD for the corresponding author in Editorial Manager on papers submitted after December 6th, 2016. Please ensure that you have an ORCID iD and that it is validated in Editorial Manager. To do this, go to ‘Update my Information’ (in the upper left-hand corner of the main menu), and click on the Fetch/Validate link next to the ORCID field. This will take you to the ORCID site and allow you to create a new iD or authenticate a pre-existing iD in Editorial Manager.

Thanks for your article submission. Please address all the comments/questions raised by reviewers.

Reviewers' comments:

Reviewer's Responses to Questions

**Comments to the Author**

1. Is the manuscript technically sound, and do the data support the conclusions?

Reviewer #1: Yes

Reviewer #2: Yes

2. Has the statistical analysis been performed appropriately and rigorously? 

Reviewer #1: I Don't Know

Reviewer #2: Yes

3. Have the authors made all data underlying the findings in their manuscript fully available?

Reviewer #1: Yes

Reviewer #2: Yes

4. Is the manuscript presented in an intelligible fashion and written in standard English?

Reviewer #1: Yes

Reviewer #2: No

5. Review Comments to the Author

Reviewer #1: This paper aims to develop and validate a predictive model of LAT/SEC risk in NVAF patients using ML methods. This paper included 1078 patients and six independent factors were identified as predictors of outcome. I have the following questions/comments:

1. Risk of stroke in patients with LAT vs. SEC are very different, SEC whilst may represent higher risk for potential development of LAT which may then lead to cardio-embolic stroke but the risk certainly would not be as high as those with established LAT. SEC sometimes may also be cleared when given isoproterenol infusion with increase in heart rates. Therfore these two outcomes should be separately observed and analyzed. And how did you analyze sludge?

2. Baseline mean LVEF 62 and 59 in those detected with LAT/SEC suggesting an overall less comorbid patient selection. What percentage of patients had reduced LVEF and heart failure? Did these patients have higher risk (and different risk factors) for LAT/SEC?

3. Patients selected from cohort undergoing catheter ablation- representing a targeted group, but it wasn't clearly stated if the following high risk groups were excluded from the study or not (they probably should be, or at least state how many patients had these): permanent AF, rheumatic heat disease, severe valve disease (especially mitral stenosis), hypertrophic cardiomyopathy, cardiac amyloidosis, patients who cannot tolerate at least 3 month of OAC post ablation.

4. How many patients were on anticoagulation at time of study and which type? And were they on long-term or only just around time of ablation? And did these factors influence risk of SEC/LAAT?

5. Did any patients undergo cardiac CT as pre imaging to ablation? Were these findings concordant with TEE? Are there any CT factors that can be incorporated into the risk model?

6. It's not quite clear which risk factors were part of which risk models. Are you able to provide maybe in appendix tables of all the parameters incorporated in each risk model and their relative weightings?

7. Add CHADSVASC to table 3 and show how it performed in the test and validation sets for predicting your outcomes compared with other models.

8. Can you suggest how you would use your findings in clinical practice? Which model would you use and in which situation. Clinically for example, if the risk is high according to your score, what further tests would you do, and what changes to management (including bloodthinner strategy) would you do?

Reviewer #2: 1- In the introduction, substitute the word "stands as" (line 21) with "is". Moreover, there is redundancy in the introductory sentence where you mention heart failure and diminishing left ventricular ejection fraction, so please adjust.

2- Also in introduction lines 24-27, merge these two sentences since the message you are sending is increasing AF prevalence with time. Simplify this part to convey your message by saying for example "the prevalence of AF is on the rise, with x numbers in 2010, that rose to x number in 2017 and is projected to double by number in the year 2050.

3- Please specify in the introduction also that left atrial appendage thrombus is part of left atrial thrombi (LAT), because it is not mentioned clearly to the reader in all of the manuscript and LAT might only evoke for some that left atrial mural thrombosis, which is part of LAT but is much less than LAA clots in terms of prevalence.

4- In the sentence discussing the role of TEE, would adopt saying that TEE is semi-invasive by nature, with rare but inherent risks and leads to patient discomfort and necessitates some form of sedation. There is a typo in the last word of this sentence (interpreted), correct to "interpretation".

5- In line 15 of second page, adjust sentence and remove the type "such", so the sentence becomes machine learning to develop predictive models for thrombus formation in AF.

6- Line 18 of the same page, delete the words most crucial feature -> the important variables

7- I congratulate the authors for developing this sophisticated ML algorithm, with the intention to convey a strong message of novelty in the filed since our only validated model to predict thrombosis and stroke risk in NVAF is the CHADVASC score. However, I would like to further explain some of the limitations and issues of this paper but not rebuke its findings. First of all, the CHADSVASC score was designed to predict risk of ischemic stroke in NVAF, but not predict the risk of LAT/SEC. There should be a modification in the objectives of the study, since comparing the predictive ability of the ML model (to detect LAT/SEC) to the CHADSVASC score is reasonable but is not 100% accurate and correct. Moreover, presence of LAT/SEC significantly increases the risk of stroke and is a contraindication for electrical cardioversion in AF, but the risk of having LAT/SEC is not equivalent to risk of developing stroke. This is a striking point that needs to be revised in the paper.

8-Would mention in exclusion criteria that patients with rheumatic HD and VHD were excluded, and those who did not have a TEE were also excluded.

9- Another important criterion that was overlooked was patients with prosthetic heart valves, since there is no mention of this population. Moreover, patients with HCM amyloidosis are excluded from stroke prediction models in AF due to their inherently increased risk and are usually anticoagulated regardless of CHADVASC score. HCM patients were included in the study, and no mention of patients with amyloid. In the latter, if data is missing or not available, please mention for the reader.

10- In terms of echocardiographic parameters, please specify to the reader that TTE data are used for the ML program. Knowing the limitations of LA diameter in 2D, why was not left atrial volume used instead? What is the utility of mitral and tricuspid regurgitation area when compared to other parameters entailing assessment of MR/TR ? What if the patient enrolled did not exhibit any MR/TR ? Please elaborate on this point because the echocardiographic model could have been markedly improved by looking at other parameters, especially since a statistically important predictor was enlarged LAD.

11- Bias control section: lines 19-20, grammatically incorrect, please make required adjustments.

12- In discussion section, the referenced study (ref 16) in the second sentence of this paragraph discussed predictability of the CHADVASC score for LAT (not SEC) only in those younger than 65 years of age. Their results do not extrapolate specifically to your study.

6. PLOS authors have the option to publish the peer review history of their article (what does this mean?). If published, this will include your full peer review and any attached files.

Reviewer #1: No

Reviewer #2: No

---

## [Author Response · Author response to Decision Letter 0]

25 Oct 2024

Dear Editor and Reviewer,

We are grateful for the opportunity to revise our manuscript titled "Development and validation of an interpretable machine learning model for predicting left atrial thrombus or spontaneous echo contrast in non-valvular atrial fibrillation patients" (Manuscript ID: PONE-D-24-27654). We appreciate the thoughtful and constructive feedback provided by you and the reviewers, which has greatly improved the quality of our work.

In this revised version, we have carefully addressed all the comments and suggestions. A brief summary of the major revisions is provided below:

1.We have revised the entire manuscript to conform to the PLOS ONE style requirements.

2.The current work was funded by the National Natural Science Foundation of China (Project No. 82070524). The funders had no role in study design, data collection and analysis, decision to publish, or preparation of the manuscript.

3.We have added the ORCID information and revised the Data Availability Statement. 

Meanwhile, we have carefully considered all your comments and made the necessary revisions to address them. Below, we provide a point-by-point response to each of your comments:

Response to Reviewer 1’s Comments:

Reviewer 1 Comment 1: Risk of stroke in patients with LAT vs. SEC are very different, SEC whilst may represent higher risk for potential development of LAT which may then lead to cardio-embolic stroke but the risk certainly would not be as high as those with established LAT. SEC sometimes may also be cleared when given isoproterenol infusion with increase in heart rates. Therfore these two outcomes should be separately observed and analyzed. And how did you analyze sludge?

Response: Thank you for pointing out this issue. We also took this into account when analyzing the data. Among all 1,078 patients, there were only 15 cases of LAT, accounting for 1.39%, and 93 cases of SEC, accounting for 8.63%. (Page 8, lines 2 to 3) There were 5 cases of sludge. Due to the small number of LAT and sludge cases, effective analysis was not feasible, so we combined LAT and SEC for analysis. As you mentioned, LAT and SEC represent different stroke risks. We have slightly adjusted the aim of the study. The goal of our study is to develop a predictive model for LAT/SEC using machine learning, with the aim of identifying AF patients at high risk for LAT/SEC. This would allow for more selective use of diagnostic tests, such as TEE. This enables targeted diagnostic evaluations and the development of personalized treatment strategies based on the findings. (Page 2, lines 16 to 18)

Reviewer 1 Comment 2: Baseline mean LVEF 62 and 59 in those detected with LAT/SEC suggesting an overall less comorbid patient selection. What percentage of patients had reduced LVEF and heart failure? Did these patients have higher risk (and different risk factors) for LAT/SEC?

Response: Thank you for raising this question. In Table 1, we grouped the LVEF data. The results show that AF patients with an LVEF <50% account for 7.98% of the total population. In the control group, 6.49% of AF patients had an LVEF <50%, whereas in the LAT/SEC group, 21.3% of AF patients had an LVEF <50%, which is significantly higher than in the control group (P < 0.001). This suggests that AF patients with an LVEF <50% have a higher risk of LAT/SEC. (Table 1)

Reviewer 1 Comment 3: Patients selected from cohort undergoing catheter ablation- representing a targeted group, but it wasn't clearly stated if the following high risk groups were excluded from the study or not (they probably should be, or at least state how many patients had these): permanent AF, rheumatic heat disease, severe valve disease (especially mitral stenosis), hypertrophic cardiomyopathy, cardiac amyloidosis, patients who cannot tolerate at least 3 month of OAC post ablation.

Response: Thank you for your valuable suggestion regarding the exclusion criteria. In response, we have revised the exclusion criteria to provide clearer and more precise definitions. Specifically, we have modified the text to state:“The exclusion criteria were as follows: (1) patients who were unable to cooperate or unwilling to participate; (2) patients with rheumatic heart disease or severe valvular heart disease; (3) those with permanent AF; (4) patients unable to tolerate at least 3 months of OAC post-ablation; (5) those who did not undergo TEE; (6) patients with incomplete clinical data.”(Page 4, lines 8 to 13). We included patients with hypertrophic cardiomyopathy and provided the relevant data. Notably, in patients with HCM or cardiac amyloidosis in conjunction with AF, the risk of stroke is significantly increased. Current studies recommend that such patients should routinely receive anticoagulation therapy, regardless of their CHA2DS2-VASc score. However, in this study, the prevalence of HCM did not differ significantly between the two groups, which may be due to the small sample size of HCM patients, leading to a lack of statistical significance. Additionally, patients with cardiac amyloidosis were not mentioned in this study due to the unavailability of relevant data. (Page 8, lines 11 to 18)

Reviewer 1 Comment 4: How many patients were on anticoagulation at time of study and which type? And were they on long-term or only just around time of ablation? And did these factors influence risk of SEC/LAAT?

Response: Thank you for raising the issue regarding the use of anticoagulant medications and their impact on LAT/SEC risk. We have considered this matter, and all patients were administered OAC treatment upon admission. However, due to a significant lack of pre-admission data, we were unable to conduct a statistical analysis on this aspect. This limitation has been acknowledged and discussed in the limitations section of our study. (Page 20, lines 3 to 5)

Reviewer 1 Comment 5: Did any patients undergo cardiac CT as pre imaging to ablation? Were these findings concordant with TEE? Are there any CT factors that can be incorporated into the risk model?

Response: Nearly all patients underwent left atrial CTA, which provided consistent diagnoses for left atrial thrombus but was unable to diagnose SEC. We did not include related parameters (such as left atrial appendage morphology, ostium size, or depth) because the primary aim of our study was to develop a machine learning model using easily accessible clinical data to identify patients at high risk of LAT/SEC. This approach allows for the targeted selection of high-risk patients for further diagnostic evaluations, such as TEE or left atrial CTA.

Reviewer 1 Comment 6: It's not quite clear which risk factors were part of which risk models. Are you able to provide maybe in appendix tables of all the parameters incorporated in each risk model and their relative weightings?

Response: Thank you for raising this question. In Fig 4a-b, we have illustrated the relative importance of each variable in the best-performing model, reflecting their relative weightings. Additionally, in the S2 Fig at the Supporting Information file, we have presented the relative weightings of each variable across all 9 models. (Page 15, lines 9 to 13 and S2 Fig)

Reviewer 1 Comment 7: Add CHADSVASC to table 3 and show how it performed in the test and validation sets for predicting your outcomes compared with other models.

Response: Thank you for raising this issue. After careful consideration, we believe it is more appropriate to include this information in Table 4. Additionally, we have presented the performance of CHA2DS2-VASc in predicting LAT/SEC across the training, testing, and validation sets. (Page 13, lines 6 to 7 and Table 4)

Reviewer 1 Comment 8: Can you suggest how you would use your findings in clinical practice? Which model would you use and in which situation. Clinically for example, if the risk is high according to your score, what further tests would you do, and what changes to management (including bloodthinner strategy) would you do?

Response: Thank you for pointing this out. The S1 Fig at the Supporting Information file presents the KS statistic plot which shows that when the absolute KS value reaches its maximum (0.598), the predicted probability is 0.104. This indicates strong model discrimination, with patients at higher risk of LAT/SEC when the predicted score exceeds 0.104. (Page 13, lines 17 to 20 and S1 Fig) Additionally, we present a representative case to illustrate the model's interpretability: an NVAF patient with LAT/SEC exhibiting a high SHAP predictive score (0.17), as depicted in Figure 4c. (Page 15, lines 13 to 15) In clinical practice, based on our newly developed predictive model, patients with NVAF who have a high SHAP prediction score—specifically, above 0.104—are at an increased risk of LAT/SEC. Clinicians should recommend that such patients undergo TEE to evaluate the presence of LAT/SEC. Additionally, comprehensive assessments of left atrial and left atrial appendage function, including parameters such as strain or fibrosis, should be performed. Finally, treatment strategies, including anticoagulation, catheter ablation, or LAA closure, should be determined based on the evaluation results. (Page 17, lines 25 to Page 18, lines 3)

Response to Reviewer 2’s Comments:

Reviewer 2 Comment 1: In the introduction, substitute the word "stands as" (line 21) with "is". Moreover, there is redundancy in the introductory sentence where you mention heart failure and diminishing left ventricular ejection fraction, so please adjust.

Response: Thank you for pointing out this error. We have corrected it to: “Atrial fibrillation (AF) is one of the most prevalent arrhythmias, predisposing individuals to thromboembolic events, heart failure, and hospitalizations, while concurrently diminishing life quality, exercise capacity.” (Page 2, lines 23 to lines 25)

Reviewer 2 Comment 2: Also in introduction lines 24-27, merge these two sentences since the message you are sending is increasing AF prevalence with time. Simplify this part to convey your message by saying for example "the prevalence of AF is on the rise, with x numbers in 2010, that rose to x number in 2017 and is projected to double by number in the year 2050.

Response: Thank you for pointing out the issue. We have changed it to: “The prevalence of AF is on the rise, with 33.5 million in 2010, that rose to 37.6 million in 2017, and is projected to double by number in the year 2050.” (Page 2, lines 25 to lines 27)

Reviewer 2 Comment 3: Please specify in the introduction also that left atrial appendage thrombus is part of left atrial thrombi (LAT), because it is not mentioned clearly to the reader in all of the manuscript and LAT might only evoke for some that left atrial mural thrombosis, which is part of LAT but is much less than LAA clots in terms of prevalence. left atrial appendage thrombus is part of left atrial thrombi (LAT).

Response: Thank you for your suggestion. We have made the following additions regarding this point: “Left atrial appendage thrombus is part of LAT. The unique anatomical features and functional properties of the left atrial appendage render it the primary site for thrombus formation. LAT and SEC are recognized as significant contributors to cardiogenic embolism in NVAF.” (Page 3, lines 3 to lines 6)

Reviewer 2 Comment 4: In the sentence discussing the role of TEE, would adopt saying that TEE is semi-invasive by nature, with rare but inherent risks and leads to patient discomfort and necessitates some form of sedation. There is a typo in the last word of this sentence (interpreted), correct to "interpretation".

Response: Thank you for pointing out the error. We have revised the sentence to: “However, transesophageal echocardiography (TEE), the gold standard for detecting LAT and SEC, is semi-invasive by nature, with rare but inherent risks and leads to patient discomfort and necessitates some form of sedation. And it demands specialized skills for accurate performance and interpretation.” (Page 3, lines 6 to lines 10)

Reviewer 2 Comment 5: In line 15 of second page, adjust sentence and remove the type "such", so the sentence becomes machine learning to develop predictive models for thrombus formation in AF.

Response: Thank you for pointing out the error. We have revised the sentence to: “there is limited research on utilizing machine learning to develop predictive models for thrombus formation in AF patients.” (Page 3, lines 16 to lines 17)

Reviewer 2 Comment 6: Line 18 of the same page, delete the words most crucial feature -> the important variables

Response: Thank you for pointing out the error. We have revised the sentence to: “Therefore, this study pursues three primary objectives: firstly, to pinpoint the important variables for predicting LAT/SEC in NVAF patients.” (Page 3, lines 18 to lines 19)

Reviewer 2 Comment 7: I congratulate the authors for developing this sophisticated ML algorithm, with the intention to convey a strong message of novelty in the filed since our only validated model to predict thrombosis and stroke risk in NVAF is the CHADVASC score. However, I would like to further explain some of the limitations and issues of this paper but not rebuke its findings. First of all, the CHADSVASC score was designed to predict risk of ischemic stroke in NVAF, but not predict the risk of LAT/SEC. There should be a modification in the objectives of the study, since comparing the predictive ability of the ML model (to detect LAT/SEC) to the CHADSVASC score is reasonable but is not 100% accurate and correct. Moreover, presence of LAT/SEC significantly increases the risk of stroke and is a contraindication for electrical cardioversion in AF, but the risk of having LAT/SEC is not equivalent to risk of developing stroke. This is a striking point that needs to be revised in the paper.

Response: Thank you for your affirmation of our research. And thank you for your suggestion. We have made appropriate adjustments to the objectives of the study. We have made the following adjustments: In the conclusion section, “The constructed logistic regression model, along with SHAP interpretation, may serve as a clinically useful tool for identifying high-risk NVAF patients. This enables targeted diagnostic evaluations and the development of personalized treatment strategies based on the findings.”(Page 2, lines 15 to lines 18) In the discussion section, “In clinical practice, based on our newly developed predictive model, patients with NVAF who have a high SHAP prediction score—specifically, above 0.104—are at an increased risk of LAT/SEC. Clinicians should recommend that such patients undergo TEE to evaluate the presence of LAT/SEC. Additionally, comprehensive assessments of left atrial and left atrial appendage function, including parameters such as strain or fibrosis, should be performed. Finally, treatment strategies, including anticoagulation, catheter ablation, or LAA closure, should be determined based on the evaluation results.” (Page 17, lines 25 to Page 18, lines 3) 

Reviewer 2 Comment 8: Would mention in exclusion criteria that patients with rheumatic HD and VHD were excluded, and those who did not have a TEE were also excluded.

Response: Thank you for your valuable suggestion regarding the exclusion criteria. We have modified the text to state:“The exclusion criteria were as follows: (1) patients who were unable to cooperate or unwilling to participate; (2) patients with rheumatic heart disease or severe valvular heart disease; (3) those with permanent AF; (4) patients unable to tolerate at least 3 months of OAC post-ablation; (5) those who did not undergo TEE; (6) patients with incomplete clinical data.”(Page 4, lines 8 to 13).

Reviewer 2 Comment 9: Another important criterion that was overlooked was patients with prosthetic heart valves, since there is no mention of this population. Moreover, patients with HCM amyloidosis are excluded from stroke prediction models in AF due to their inherently increased risk and are usually anticoagulated regardless of CHADVASC score. HCM patients were included in the study, and no mention of patients with amyloid. In the latter, if data is missing or not available, please mention for the reader.

Response: Thank you for pointing this out. We included patients with hypert

---

## [Editor Report · Decision Letter 1]

28 Oct 2024

Development and validation of an interpretable machine learning model for predicting left atrial thrombus or spontaneous echo contrast in non- valvular atrial fibrillation patients

PONE-D-24-27654R1

Dear Dr. Li,

We’re pleased to inform you that your manuscript has been judged scientifically suitable for publication and will be formally accepted for publication once it meets all outstanding technical requirements.

Kind regards,

Tom Wang

Academic Editor

PLOS ONE

Additional Editor Comments (optional):

Thanks for addressing all the reviewer's comments adequately.

---

## [Editor Report · Acceptance letter]

25 Nov 2024

PONE-D-24-27654R1 

PLOS ONE

Dear Dr. Li, 

I'm pleased to inform you that your manuscript has been deemed suitable for publication in PLOS ONE. Congratulations! Your manuscript is now being handed over to our production team.

Kind regards, 

on behalf of

Dr. Tom Kai Ming Wang 

Academic Editor

PLOS ONE